# Sustainable Treatment and Resource Recovery of Anion Exchange Spent Brine by Pilot-Scale Electrodialysis and Ultrafiltration

**DOI:** 10.3390/membranes12030273

**Published:** 2022-02-27

**Authors:** Hongfang Sun, Daoxu Zhu, Peng Shi, Wenxiang Ji, Xun Cao, Shi Cheng, Yufeng Lou, Aimin Li

**Affiliations:** 1State Key Laboratory of Pollution Control and Resources Reuse, School of the Environment, Nanjing University, Nanjing 210023, China; shf4514356@163.com (H.S.); 18851777819@163.com (D.Z.); shipeng@nju.edu.cn (P.S.); jwx1996_nju@163.com (W.J.); shicheng_nju@163.com (S.C.); 2Yancheng Academy of Environmental Protection Technology and Engineering, Nanjing University, Yancheng 224000, China; caoxun890310@163.com; 3Key Laboratory of Charged Polymeric Membrane Materials of Shandong Province, Shandong Tianwei Membrane Technology Co., Ltd., 13173 Yuqing East Street, Weifang 261061, China; yf_lou@163.com; 4Quanzhou Institute for Environmental Protection Industry, Nanjing University, Quanzhou 362008, China

**Keywords:** ED + UF hybrid process, AIX spent brine, NaCl recovery, resin regeneration, HS liquid fertilizer

## Abstract

The anion exchange (AIX) spent brine, generated during the NDMP-3 resin regeneration process, highly loaded with organic substances mainly humic substances (HSs) and salts (mainly NaCl) remains an environmental concern. In this study, pilot-scale electro dialysis (ED) and ultrafiltration (UF) hybrid technologies were first used to recover NaCl solution as a resin regeneration agent and HSs, which could be utilized as a vital ingredient of organic fertilizer, from the AIX spent brine. Recovered ≈ 15% *w*/*w* NaCl solution obtained by two-stage pilot-scale ED can be used to regenerate saturated NDMP-3 anion exchange resins; the regeneration–readsorption performance of NDMP-3 resins was equivalent to that of fresh ≈ 15% *w*/*w* NaCl solution. The two-stage dilute solution with low-salt content (0.49% *w*/*w*) was further concentrated by pilot-scale UF, so that the HS content in the retentate solution was >30 g/L, which meets the HS content required for water-soluble organic fertilizers. The HS liquid fertilizer could significantly stimulate the growth of green vegetables with no phytotoxicity, mainly due to special properties of HSs. These results demonstrate that ED + UF hybrid technologies can be a promising approach for the sustainable treatment and resource recovery of AIX spent brine.

## 1. Introduction

Anion exchange (AIX) as advanced drinking water treatment technology that can effectively remove natural organic matter (NOM) in raw water and control disinfection by product formation [1,2,3]. Nevertheless, AIX spent brine generated during the anion exchange resin regeneration process is a main bottleneck problem [4,5], which contains the excess NaCl of regeneration solution and desorbed high levels of organic matters, mainly humic substances (HSs) and some inorganic salts from raw water [3]. This spent brine has high conductivity and poor biodegradability, because humic substances are biorefractory compounds [6]. The high salinity in the AIX spent brine could also harm the biodegradation efficacy. Several methods to conventionally dispose of the AIX spent brine, such as discharge after dilution, deep well injection, landfill, discharge in the sewer or other surface water bodies, sea and ground storage, and evaporation pounds, have been investigated [7,8,9]. However, these methods either cause secondary pollution such as severe groundwater, surface water, soil and sea contamination, causing the destruction of marine ecological balance and soil salinization [3,4,10], or high energy consumption and operation costs [11].

For sustainable wastewater treatment, resource recovery from the spent brine is a significant research direction [12]. It is reported that HSs, as an important component of organic fertilizer, are beneficial to plant growth [12,13]. The AIX spent brine contains plenty of humic substances. Nevertheless, its characteristics of high inorganic salts and low HS content limit its direct application as an organic fertilizer. Therefore, it is vital to separate the inorganic salts (mainly NaCl) and HSs in the AIX spent brine. NaCl and HSs are then reused as valuable secondary products, respectively. The separated and concentrated high-concentration NaCl solution can be recycled and reused as a saturated resin regeneration agent to reduce the use of industrial sodium chloride in the ion exchange process [3,13]. In addition, concentrations of separated HS are also necessary in order to obtain high concentration of HS as an organic fertilizer.

Electrodialysis (ED) is an effective electro-membrane separation technology for inorganic salts and organic substances, utilizing the permselectivity of ion exchange membranes to achieve separation [14,15]. ED technology could be applied in the concentration of seawater to produce salts, concentration and attainment of salts from seawater reverse osmosis concentrates, and brackish water or wastewater desalination [16,17,18,19]. In recent years, desalination of AIX spent brine with ED is a research trend. In one study, effective separation of salts and NOM in the synthetic AIX spent brine with pressure-driven (nanofiltration and ultrafiltration) and electric-driven (ED) technologies was described [20], but simulated solution of humic acid (HA),produced by Aldrich, with marked different characteristics compared to aquatic NOM solution was used that cannot represent the real AIX spent brine. Furthermore, the resource utilization study of NaCl and HA after separation was not systematically discussed. In another study, recovering NaCl and NOM in the spent brine was studied by two-stage ED [13]. Nevertheless, the recovered NaCl solution concentration cannot meet the concentration requirement as a saturated resin regeneration agent, and resource reuse of recovered NaCl and NOM was not investigated. Additionally, desalination performance of ion exchange spent brine via ED, about applied electric field and the ion exchange membrane permselectivity impact, was discussed [11]. However, the practical application of the recovered NaCl solutions and transport of organic matters during the ED process were not systematically evaluated. Moreover, the fate and disposal method of the high concentration recovered NOM was also not illustrated.

Ultrafiltration (UF) using tight membranes, whose molecular weight cutoff ranges is located at approximately 1000–10,000 Da, was reported to extract and concentrate organic matters (mainly HSs) from the membrane bioreactor plus nanofiltration (MBR + NF) leachate concentrates [21]. This might be due to the fact that the tight UF membranes could effectively retain organic matter, and the permeability of multivalent inorganic ions is somewhat high. Thus far, HSs’ concentration and recovery via UF process from AIX spent brine has never been investigated. Furthermore, no references have been found regarding the production of value-added humic substance organic fertilizers from AIX spent brine to cultivate green vegetables.

In this study, utilizing two-stage pilot-scale ED to separate the AIX spent brine into a high concentration NaCl solution (15% *w*/*w*) and humic substance solution was firstly conducted, and the migration of organic matters was systematically studied by the size exclusion chromatography–diode array detector–fluorescence detector–organic carbon detector (SEC–DAD–FLD–OCD) system. Recovered NaCl solution, that is one-stage concentrate solution of ED, was then used to regenerate saturated NDMP-3 anion exchange resins. We then proceeded with, discussing NDMP-3 resins’ regeneration–readsorption performance regenerated by the recovered NaCl solution, compared to that of fresh NaCl solution. Secondly, the further concentration of the HS solution, that is two-stage dilute solution of ED, was studied via the pilot-scale ultrafiltration under different influent flow in order to harvest the HS of specific concentration as an organic fertilizer that meets industry standards. Finally, the pilot test on cultivation of green vegetables was carried out to study the recovered HS solution’ application effect as a water-soluble organic fertilizer. The aim of this research is to verify the feasibility of using pilot-scale electrodialysis and ultrafiltration hybrid technologies for recovering NaCl solution as a resin regeneration agent and HS solution as a liquid organic fertilizer from the AIX spent brine, and to stimulate these technologies’ full-scale industrial application for sustainable treatment and resource recovery of the AIX spent brine with high-salinity and high-concentration humic substances.

## 2. Materials and Methods

### 2.1. The Anion Exchange Spent Brine

Our research group synthesized and applied the NDMP-3 anion exchange resins for the advanced treatment of drinking water [22]. The full-scale resin adsorption project (25,000 ton/d) using anion an exchange fluidized bed was built in Yancheng Dafeng Water Systems Co., Ltd. (Yancheng, China). Sand-filtered water was treated by the NDMP-3 anion exchange resins in the anion exchange fluidized bed (Figure 1). When the adsorption of resins was saturated after one week of operation, it was necessary to regenerate with a fresh 15% (*w*/*w*) NaCl solution to achieve cyclic adsorption [2]. Wastewater from the resin regeneration process was AIX spent brine. Before the pilot-scale ED experiments, vacuum filtration was conducted to filter out solid particles in the AIX spent brine.

The freeze-dried method was used to make the AIX spent brine into powder. Then Fourier transform-infrared spectroscopy (FT-IR) and thermogravimetric analysis (TGA) was carried out. The previous study discussed the other nature of AIX spent brine [3]. Fourier transform-infrared spectroscopy was conducted on a Nicolet Nexus 870 spectrophotometer using KBr powder. The thermogravimetric analysis was performed on a thermal analyzer SETSYS made by the France company, SETARAM. A sample of 40 mg powder containing AIX spent brine was put on the quartz crucible, and then it was heated until 1000 °C in helium gas (10 mL/min flow).

### 2.2. Separation Test by Two-Stage Pilot-Scale ED 

The separation performance of the AIX spent brine via two-stage pilot-scale ED was investigated with the ED pilot plant, in which the membrane size of TWEDC1 (code for anion-exchange membrane) and TWEDA1 (code for cation-exchange membrane) membrane pair (Appendix A) from Shandong Tianwei Membrane Technology Co., Ltd. was 400 mm × 800 mm, shown in Appendix A. The suitable experimental conditions previously studied in the ED process were obtained [3]. Detailed experimental conditions of two-stage pilot-scale ED are available in the Appendix A. The SEC–DAD–FLD–OCD systems [23,24], which are described in detail in the Appendix A, were used to measure the properties of AIX spent brine, second-stage dilute solution, first-stage concentrate solution, and second-stage concentrate solution to study the transport of organic matters through the ion exchange membranes (IEMs) in the two-stage pilot-scale ED process. Dilute solution conductivity and concentrate solution conductivity were determined in the pilot-scale ED experiment process. The desalination rate (*R*, %) [17] was calculated by Equation (1):*R* = (*κ*_0_ − *κ*_t_)/*κ*_0_ × 100%(1)
where *κ*_t_ and *κ*_0_ are the final and initial conductivity of the dilute solution, respectively.

Figure 1 shows the flowchart of the methodology about sustainable treatment and resource recovery of AIX spent brine applied in this study. AIX by NDMP-3 anion exchange resins treat sand-filtered water from Yancheng Dafeng Water Systems Co., Ltd.to remove NOM before disinfection. After one week of operation, NDMP-3 resins were separated and a fresh 15% (*w*/*w*) NaCl solution was used to regenerate, then regenerated resins were reused. The AIX spent brine, contains excessive NaCl and the desorbed NOM mainly HSs, but also a small amount of other anions from the raw water of the Tongyu River that could have an affinity for the resins: sulfate, nitrate, carbonate and bicarbonate [3]. To separate and recycle HSs and NaCl, two-stage pilot-scale ED (Appendix A) experiments of AIX spent brine were first conducted. Then, recovered NaCl solution that was a one-stage concentrate solution of ED compared to fresh NaCl solution, was utilized to regenerate saturated NDMP-3 anion exchange resins, and the adsorption test of NDMP-3 resins after regeneration was also investigated. Secondly, recovered HS low-salt solution, that is the two-stage dilute solution of ED, was further concentrated by pilot-scale UF (Appendix A) under different influent flow in order to produce high concentration HS solution that meets the standard of a water-soluble fertilizer. Finally, the retentate solution of UF pilot, that is the recovered HS solution, was used as the HS liquid fertilizer, then it was applied to carry out the pilot test on the cultivation of green vegetables (Appendix A) to study the application efficiency as an HS liquid fertilizer (Appendix A) compared with conventional fertilizer (Appendix A). The second-stage concentrate solution of the pilot-scale ED and permeate solution of pilot-scale UF could also be used as the initial concentrate solution in the pilot-scale ED until its NaCl content was concentrated to reach the content of 15% *w*/*w* for the saturated NDMP-3 anion exchange resin regeneration.

### 2.3. Resin Regeneration Test of Recovered NaCl Solution

The 50 mL concentrate solution produced via one-stage pilot-scale ED, which is recovered NaCl solution (≈15% *w*/*w*), was used to conduct desorption process of saturated NDMP-3 resin (50 mL) compared with the 50 mL fresh≈15% (*w*/*w*) NaCl solution. The resins were mixed with NaCl regeneration solution, and the contact time was 30 min. The permanganate index (COD_Mn_) and UV_254_ (UV absorbance at 254 nm) of two AIX spent brine were determined to study the regeneration effect of recovered NaCl solution and fresh NaCl solution. 1 mL NDMP-3 resins after the regeneration of recovered NaCl solution and fresh NaCl solution were utilized to adsorb NOM mainly humic substances in 3600 mL sand-filtered water from Yancheng Dafeng Water Systems Co., Ltd. for the purpose of discussing ion exchange performance. Under a simulated suspended bed operation, each adsorption was performed in batches with 300-bed volume (BV, the ratio of resin volume to water volume) until 3600 BV and the contact time of each adsorption was 30 min. The UV_254_ and dissolved organic carbon (DOC) removals of sand-filtered water after each adsorption were used to investigate the adsorption behavior of NDMP-3 resins regenerated by recovered NaCl solution and fresh NaCl solution.

### 2.4. Pilot-Scale UF Test for HS Concentration

A pilot-scale UF (Appendix A) test was adopted to further concentrate the two-stage dilute solution of the ED pilot in order to generate a retentate solution of UF in which the content of HS meets the standard as a liquid fertilizer. A roll-type tight UF membrane (molecular weight cut-off 3000 Da, membrane area 30.4 m^2^) was kindly supplied by Trans film (North Hollywood, CA, USA). Cross-flow filtration of UF pilot was employed. The inlet water pressure did not exceed 1.2 MPa, and the difference between membrane pressure and retentate solution outlet pressure was operated on within 0.1–0.15 MPa. The influence of influent flow on the concentration efficiency in the UF pilot was investigated at 16.9, 19.1, 24.4, 25.4 and 27.9 L/min. Water was used to clean the membrane for flux recovery.

Volume concentration factor (*CF*) in the pilot-scale UF was determined based on Equation (2):*CF* = *V_f_*/*V_r_*(2)
where *V_f_* and *V_r_* are volumes of the feed and the retentate solution, respectively.

Permeation flux (*J*) and rejection coefficient (*R*) of the UF membrane for the pilot-scale UF process were calculated by Equations (3) and (4), respectively.
*J* = *V*/(*A* × *t*)(3)
*R* = (*C_f_* − *C_p_*)/*C_f_* × 100%(4)
where *V* is the volume of permeate solution at operation time *t*, and *A* is the effective UF membrane area. *C_f_* and *C_p_* are the COD_Mn_ concentrations in the feed and permeate solution, respectively.

### 2.5. Pilot Test on the Cultivation of Green Vegetables with HS Liquid Fertilizer

The HS liquid fertilizer was made up by adding 127.8 g/L urea, 69.6 g/L potassium dihydrogen phosphate and 137.7 g/L potassium nitrate to the retentate solution of the UF pilot, which acted as a macro-element HS fertilizer, and its composition is shown in Appendix A. Conventional fertilizer (Appendix A) and 46 L HS fertilizer mixed respectively with 6.03 kg urea, 2.5 kg calcium superphosphate, and 5.3 kg K_2_SO_4_ were used as a base fertilizer (fertilization one time) and a fertilizer for top dressing (fertilization three times), then they were used to cultivate the green vegetables and compare fertilizer efficiency.

A plot of 7 m^2^ divided into 6 fields was used to plant green vegetables. A total of 350 green vegetables were cultivated in each field. The green vegetables were planted at a distance of 20 cm × 20 cm. Normal pest control and plant management were performed throughout the growth period of the plants. After 20 days, green vegetables were picked to determine the fresh weight, dry weight, plant height and relative content of chlorophyll. Then, green vegetables were top-dressed three times. After one month, the green vegetables were harvested, and the total nitrogen, total potassium, crude fiber, and soluble sugar content of the green vegetables were measured to examine the quality of the green vegetables. Ten vegetables were continuously picked up, roots were cut off, washed and dried, to measure the fresh weight. Then dried at 60° to a constant weight in an oven at 105° for 15 min to measure the dry weight of the vegetables. The height of the plant was obtained from the height of the ground part. A chlorophyll meter (SPAD-502Plus) was used to determine the relative content of chlorophyll. The fully stretched four plants of leaves were obtained, and measured at 3 points on each leaf. The determination of total nitrogen, total potassium, crude fiber and soluble sugar content in green vegetables was entrusted to a testing company.

## 3. Results and Discussion

### 3.1. AIX Spent Brine Characterization with FT-IR and TGA

AIX spent brine was alkaline wastewater with high turbidity, hydrophilicity, high aromaticity, abundant oxygen-containing functional groups, high concentration salts (mainly NaCl) and organic matters (mainly HSs) according to our previous research study [3]. The FT-IR spectrum, the thermogravimetry (TG), and derivative thermogravimetry (DTG) profiles shown in Figure 2a,b were conducted to further study the functional groups and thermal properties of AIX spent brine.

The FT-IR spectrum in Figure 2a indicated the bands located at 3435, 2920, 2852, 1621, 1394, 1143, 997,834, 703 and 621 cm^−1^ displaying an absorption spectrum with typical HS bands. The H-bonded O−H stretching of carboxylic acids, phenols, alcohols or H-bonded N-H stretching were located at the broad band of 3435 cm^−1^.The band at 3000–3100 cm^−1^ ascribed to =C−H groups in the aromatic rings did not exist, indicating that aromatic rings were highly substituted [25]. It was followed by a doublet at 2920 cm^−1^ and 2852 cm^−1^ corresponding to symmetric and asymmetric aliphatic C−H stretching in CH_3_ and CH_2_ groups [26]. The band at around 1621 cm^−1^ was due to the stretching of C=C vibrations in the aromatic and olefinic, and conjugated carbonyl groups’ C=O stretching in ketone, quinone and amide [27]. A weak band at 1394 cm^−1^ implied the O−H deformation, C−O stretching vibration of phenolic OH groups, C=O symmetric stretching vibration (COO^−^), and C−H deformation of aliphatic and CH_3_ groups [28,29]. At 1143 cm^−1^ and 997 cm^−1^, these two bands indicated the C−O stretching vibration in the polysaccharides or polysaccharide-like structures, phenols, alcohols and aliphatic ethers [28,29,30]. Bands at 834, 703 and 621 cm^−1^ could be reasonably attributed to the O−H stretching vibration of carboxylic groups, C=C bending and the C−H stretching vibration of substituted aromatic groups [31]. These results demonstrated that the HSs in the AIX spent brine had a higher proportion of aromatic rings, COO^−^ groups and phenolic OH groups, consistent with the data of oxygen-containing functional groups reported in the previous research [3].

From the thermogravimetric analysis in Figure 2b, it can be seen that the ash content of the AIX spent brine was 77.0%, further confirming its high salt content [3,32]. Moreover, the TG and DTG curves showed the typical characteristics of the HS thermal decomposition process [33,34,35]. Thermal decomposition (Figure 2b) for AIX spent brine possibly went through four main stages: (1) the evaporation of water, which is incorporated in or adsorbed onto the HSs, between 51 °C and 217 °C (1.5% weight loss); (2) decomposition of carbohydrates and the loss of carboxyl, methyl, methylene and alcohol groups and unsaturation loss between 217 °C and 486 °C (3.1% weight loss); (3) the loss of phenolic acid groups, carbon chain oxidation, and the condensation polymerization of aromatic structure between 486 °C and 689 °C (1.4% weight loss); and (4) the pyrolysis of aromatic structure to form coke between 689 °C and 997 °C (17% weight loss) [34,35,36]. The loss of carboxyl, alcohol groups, and phenolic acid groups in the second and third stage of thermal decomposition further verified that the HSs in the AIX spent brine had ample oxygen-containing functional groups, including carboxyl and phenolic hydroxyl groups, which is in agreement with the FT-IR spectrum results and the measurement data of oxygen-containing functional groups [3].

### 3.2. Separation Effect of NaCl and HSsin theTwo-Stage Pilot-Scale ED

Separating the AIX spent brine into NaCl and HSs was conducted by two-stage pilot-scale ED. The separated NaCl solution (≈15% *w*/*w*) was applied as a regeneration agent of saturated NDMP-3 resin, while separated HS solution can be further concentrated by pilot-scale UF, then it could be used as a HS liquid fertilizer in order to cultivate green vegetables. The suitable hydrodynamic conditions of ED have been systematically studied in the previous study [3]. The conductivity changes of AIX spent brine shown in Figure 3 during the two-stage pilot-scale ED process were obtained by using suitable reaction conditions. 

As can be seen from Figure 3a, the inorganic salts, mainly NaCl, were transported from the dilute to the concentrate solution during the first-stage pilot-scale ED process, so the conductivity of the dilute chamber continued to decrease with time, from 115.80 mS/cm to 10.82 mS/cm, while the conductivity of the concentrate chamber increased with time, from 0.73 mS/cm to 167.72 mS/cm (Table 1). The NaCl content was 14.92% *w*/*w* (≈15% *w*/*w*) in the first-stage concentrate solution that could meet the NaCl concentration requirement of saturated anion exchange resin (NDMP-3) regeneration. Desalination rate and COD_Mn_ retention of the first-stage dilute solution were respectively 90.7% and 95.8% (Table 1) through first-stage pilot-scale ED. In order to reduce the salinity of the recovered HS solution as much as possible, and meet the industry standard requirement for the salinity of HS fertilizer, the first-stage dilute solution was subjected to second-stage ED for further desalination. The conductivity of the first-stage dilute solution dropped from 10.82 mS/cm to 6.09 mS/cm, while the conductivity of the second-stage concentrate solution rose from 7.12 mS/cm to 21.64 mS/cm (Figure 3b and Table 1). The second-stage concentrate solution continued to be used as the initial concentrate solution until its NaCl content reached the content of 15% *w*/*w*. It could also be used as a regeneration agent of resins. 

After the two-stage pilot-scale ED desalination, the second-stage dilute solution, that is the recovered HS solution, its NaCl content was 0.49% *w*/*w*, which meets the requirements of liquid fertilizer salinity according to Chinese professional standard NY1110-2010, and the total desalination rate and total COD_Mn_ retention of AIX spent brine were 94.7% and 95.5% (Table 1), respectively. Furthermore, it was found that ED could concentrate HSs to a certain extent by comparing the COD_Mn_ values of AIX spent brine, first-stage dilute solution and second-stage dilute solution. These results indicated that the two-stage pilot-scale ED can realize the effective separation of NaCl and HS in the AIX spent brine.

Additionally, the pH values of AIX spent brine, first-stage concentrate solution, first-stage dilute solution, second-stage concentrate solution, and second-stage dilute solution have not changed much, and were around 8; this might be because no serious water splitting occurred owing to the fact that the limiting current density was not exceeded, or there was no notable migration of H^+^ and OH^−^ during the two-stage pilot-scale ED process [17,37]. Moreover, the COD_Mn_ values of the first-stage concentrate solution and second-stage concentrate solution were 241.1 mg/L and 36.1 mg/L (Table 1), because there was a small amount of organic matter that migrated from the dilute chamber to the concentrate chamber. It is very interesting and necessary to study the migration of organic matters in the ED process to better understand what kinds of organic matters could permeate across the IEMs and whether the organic matter in the recovered NaCl solution (≈15% *w*/*w*) had an impact on the regeneration of saturated NDMP-3 resins.

The SEC–DAD–FLD–OCD analyses of AIX spent brine, first-stage concentrate solution, second-stage dilute solution and second-stage concentrate solution to study the transport of organic substances through IEMs during the two-stage pilot-scale ED process are shown in Figure 4. According to previous research [23,24], the DAD measurements (the wavelength range > 240 nm) characterize components that respond to UV absorbance, especially the aromatic and unsaturated functional groups in NOM. The FLD detector with multiple emission wavelength scanning identified humic-like substances at Em > 380 nm and protein-like substances at Em < 380 nm [38]. Organic carbon was responded by the OCD detector. Based on the SEC–DAD–FLD–OCD fingerprints and fraction assignation of surface water samples studied by Huber et al. [24], the chromatograms of AIX spent brine, first-stage concentrate solution, second-stage dilute solution and second-stage concentrate solution were classified into four major fractions, including biopolymers (BP), humic substances (HS), building blocks (BB) of humic substances and low molecular weight substances (LMWS) in this work. 

The organic matters in the AIX spent brine were mainly HSs (>80%) [3,4]. They exhibited strong humic-like fluorescence peaks in the FLD chromatogram and intensive UV absorbance at wavelengths >240 nmin the DAD chromatogram, shown in Figure 4a. This reflected the presence of aromatic moieties typical for HSs [23]. The peak value of HSs was<600 mg/Lin the OCD chromatogram, for which apparent molecular weight (AMW) was located at ~7.2 kDa to ~1.3 kDa. In addition, AIX spent brine contained a small quantity of biopolymers (the AMW was >20 kDa), building blocks of humic substances (the AMW located at ~1.3 kDa to ~500 Da), and low molecular weight substances (the AMW was <500 Da) [4]. BP has been reported as mainly being polysaccharides, acromolecular proteins and amino acids [24]. The molecular weight of BB was relatively small, and it might be degradation products of humic substances as well as made up of both aromatic and aliphatic low molecular weight acids [23,24]. LMWS was mainly composed of low molecular weight neutrals. It had low aromaticity and did not contain humus-like fluorescence, and might be mainly composed of small molecule substances such as amino acids, fatty acids, alcohols, aldehydes, ketones sugars [39,40]. However, nitrate also exhibited the notable UV absorbance at wavelength <240 nm in the DAD chromatograms due to π →π* bands absorbance, and nitrate and nitrite both have UV absorbance at wavelength >250 nm because of n →π* bands absorbance, so the characterization of low molecular weight neutrals by UV absorbance (such as UVA254) was complicated due to the interferences of nitrate and nitrite.

Second-stage dilute solution (Figure 4c) showed stronger humic-like fluorescence peaks in the FLD chromatogram and stronger UV absorbance at wavelengths >240 nmin the DAD chromatogram than that of AIX spent brine, and the peak value of HSs (the apparent molecular weight AMW located at ~7.2 kDa to ~1.3 kDa) was ≈600 mg/Lin the OCD chromatogram, suggesting that it had a higher content of HSs than AIX spent brine. This further confirmed HSs were concentrated by two-stage pilot-scale ED, which was in agreement with COD_Mn_ measurement results in Table 1. Moreover, second-stage dilute solution, in which the AMW was <500 Da, exhibited the more notable UV absorbance at wavelength <240 nm in the DAD chromatogram, indicating the low molecular weight neutrals and nitrate or nitrite were also concentrated by two-stage pilot-scale ED. 

Figure 4b shows that the first-stage concentrate solution had some HSs, BB of humic substances, LMWS and nitrate or nitrite, in which the peak value of HSs (the AMW located at ~7.2 kDa to ~1.3 kDa) was >25 mg/L, the peak value of BB (the AMW located at ~1.3 kDa to ~500 Da)was >5 mg/L and the peak value of LMWS and nitrate or nitrite (the AMW located at <500 Da) was >10 mg/Lin the OCD chromatogram. These results indicated that some negatively charged and hydrophilic HSs and BB, and low molecular weight hydrophilic neutrals and nitrate or nitrite could migrate through IEMs from the dilute to the concentrate solution, consistent with the research results reported in the literature [3,41]. So the charge and molecular size of organic substances can evidently affect their transport during the ED process. 

Second-stage concentrate solution (Figure 4d) showed the similar SEC–DAD–FLD spectral features as that of first-stage concentrate solution, indicating the second-stage concentrate solution also had some HSs, BB of humic substances, LMWS and nitrate or nitrite. However, it exhibited stronger humic-like fluorescence peaks in the FLD chromatogram and stronger UV absorbance at wavelengths >240 nmin the DAD chromatogram than that of first-stage concentrate solution. These suggested that the second-stage concentrate solution had more HSs with UV absorbance and the humic-like fluorescence absorbance functional group, but the HS content was lower than that of first-stage concentrate solution because of the lower peak value(<20 mg/L) of HSs (the AMW located at ~7.2 kDa to ~1.3 kDa) in the OCD chromatogram. In addition, according to the DAD and OCD chromatograms, the second-stage concentrate solution had more humus-like fluorescence BB (the AMW located at ~1.3 kDa to ~500 Da), but the total content of BB was lower than that of the first-stage concentrate solution. Moreover, the content of LMWS and nitrate or nitrite (the AMW located at < 500 Da) in the second-stage concentrate solution was also lower than that of the first-stage concentrate solution because of weaker UV absorbance at wavelength <240 nm and lower DOC content. These results indicated organic matters migrated more in the first-stage ED process than that of the second-stage ED, and the nature of organic matter migration was not the same in the first-stage ED and second-stage ED processes, which might be related to the NaCl concentration, the running time of ED and organic matter adsorption [41].

As can be seen from Appendix A, the permselectivity of TWEDC1/TWEDA1 membrane pair was ≥96%, so it cannot selectively permeate only monovalent ions (Na^+^ and Cl^−^). Nitrate or nitrite could also migrate through IEMs because they are also monovalent ions with a small hydration radius. In addition, some organic matters with the charged functional groups (HSs and BB), which are negatively charged, can also occur with the transmembrane electric migration under the electric field because of the ion exchange during the ED process (Figure 4b,d). Meanwhile, the zeta potential of TWEDA1 anion exchange membrane versus pH value was shown in Appendix A, the zeta potential of TWEDA1 anion exchange membrane was −3.5 mV at pH = 8.27, while the pH of AIX spent brine was 8.27 (Table 1), suggesting the TWEDA1 anion exchange membrane was negatively charged. So, the transport of organic substances might also be related to membrane fouling because of electrostatic interaction between TWEDA1 anion exchange membranes and negatively charged organic matters [41]. In addition, Jaeweon Cho et al. [41] reported that the major apparent pore size distribution of the CMX anion-exchange membrane and AMX cation-exchange membrane (NEOSEPTA, Japan) was located in the range of 100–200 mass units. According to our previous research results [3], the COD_Mn_ retention for the TWEDC1/TWEDA1 membrane pair with long-term running lab-scale electrodialysis was lower than that of the CM-1/AM-2 membrane pair from NEOSEPTA. It can be inferred that the pore size of the TWEDC1/TWEDA1 membrane pair was larger than that of the CMX anion-exchange membrane and AMX cation-exchange membrane. So, low molecular weight HSs and BB and low molecular weight uncharged neutrals (Figure 4b,d) could be easier to migrate through the TWEDC1/TWEDA1 membrane pair from the dilute to the concentrate solution [3]. Moreover, the contact angles of TWEDC1/TWEDA1 membrane pair were <90 degrees (Appendix A), so TWEDC1/TWEDA1 membrane pair was hydrophilic. Jaeweon Cho et al. [41] found that the hydrophilic NOM was migrated easily into the concentration chamber compared with hydrophobic and transphilic NOM fractions. So some hydrophilic HSs, BB and low molecular weight hydrophilic neutrals could transport through TWEDC1/TWEDA1 membrane pair from the dilute to the concentrate solution (Figure 4b,d), because of hydrophilic and hydrophobic effects between the hydrophilic TWEDC1/TWEDA1 membrane pair and the hydrophilic organic matters. Thus, the charge, molecular size, and hydrophilic properties of organic matter as well as the ion exchange membrane properties significantly affect the transport of organic substances during the ED process.

### 3.3. Resin Regeneration Performance with Recovered NaCl Solution 

In order to study the regeneration effect of recovered ≈ 15% (*w*/*w*) NaCl solution obtained via pilot-scale ED, it was used to regenerate the saturated NDMP-3 anion exchange resins. Fresh ≈ 15% (*w*/*w*) NaCl solution was used to compare. The UV_254_ absorbance and COD_Mn_ of two AIX spent brine regenerated by recovered NaCl solution and fresh NaCl solution are shown in Figure 5a. In the ED desalination process, owing to the permselectivity restriction of TWEDC1/TWEDA1 membrane pair [3], a small amount of organic substances including negatively charged hydrophilic HSs and building blocks, and low molecular weight hydrophilic neutrals in the dilute chamber can penetrate across the ion exchange membranes from dilute to concentrate solution (Figure 4), so the recovered NaCl solution had some organic substances, mainly HSs [3,11,13,41]. Subtracting the UV_254_ absorbance and COD_Mn_ of organic substances already existing in the recovered NaCl solution, the UV_254_ absorbance and COD_Mn_ of desorbed organic substances in the AIX spent brine regenerated by recovered NaCl solution were 151.2 cm^−1^ and 4000 mg/L (Figure 5a), respectively. These values were almost equal to the UV_254_ absorbance and COD_Mn_ of AIX spent brine regenerated by fresh NaCl solution (144.2 cm^−1^ and 3950 mg/L). An interpretation of these results is that the organic substances in the recovered NaCl solution did not affect the regeneration performance of the saturated NDMP-3 resins; because the concentration of NaCl is much higher than the concentration of residual organic matters, there is no competitive ion exchange effect [1,42]. So, the organic substances (mainly HSs) adsorbed by the NDMP-3 resin could be effectively desorbed with recovered NaCl solution.

For the purpose of further verifying the feasibility of the recovered ≈15% (*w*/*w*) NaCl solution as are generation agent of NDMP-3 resins, the readsorption behavior of NDMP-3 resins regenerated by fresh NaCl solution and recovered NaCl solution was investigated and compared. The organic substances, which were mainly HSs in the sand-filtered water, were adsorbed and removed by regenerated NDMP-3 resins. The DOC and UV_254_ absorbance removals of two kinds of NDMP-3 resins regenerated by fresh NaCl solution and recovered NaCl solution are shown in Figure 5b,c. It can be seen from Figure 5b that the removal rates of DOC by the two kinds of regenerated NDMP-3 resins were both in the range of 43–22%. The UV_254_ absorbance removal rates were both in the range of 47–27% (Figure 5c). These results implied the NDMP-3 resins regenerated by fresh NaCl solution and recovered NaCl solution had almost the same efficiency of adsorbing organic matter in the sand-filtered water. This is in agreement with the results in the literature, that the reused brine did not greatly affect the NOM removal [43]. Therefore, the recovered ≈15% (*w*/*w*) NaCl solution via pilot-scale ED is appropriate for regenerating the saturated NDMP-3 resins without the decrease in the adsorption performance of the regenerated NDMP-3 resins. The NaCl solution separated by ED can be reused effectively for resin regeneration.

### 3.4. HS Concentration by Pilot-Scale UF

The ED second-stage dilute solution with the low-salt content (0.49% *w*/*w* NaCl), that is recovered HS solution, was further concentrated by pilot-scale UF to meet the HS content required by liquid fertilizer. Figure 6 presents the effect of influent flow onCOD_Mn_ of retentate solution in the pilot-scale UF. The COD_Mn_ of retentate solution reached the maximum, that was 13,722.6 mg/L (COD_Cr_ was about 40,000 mg/L), when the influent flow was 16.9 L/min. The HS content in the retentate solution of UF pilot was >30 g/L, which meets the HS content required in Chinese professional standard NY1106-2010 for water-soluble fertilizers [21]. In addition, the membrane permeate flux decreased with increasing the volume concentration factor (CF) in the pilot UF, shown in Appendix A, which was in accordance with the result reported in the literature [21]. This might be ascribed to the enhanced resistance of the membrane because the membrane surface and pores could be retained and adsorb high concentration HSs, and diminish the electrostatic repulsion between HSs and membrane surface because high ionic strength sharply reduced the membrane and HSs surface charges [12,21,44]. The permeate solution of the pilot-scale UF can also be used as the initial concentrate solution in the pilot-scale ED, until its NaCl content was concentrated to reach the content of 15% *w*/*w* by ED as a saturated NDMP-3 anion exchange resin regeneration agent.

Some properties of the recovered HS solution originating from the retentate solution in the UF pilot process are shown in Table 2. Only trace concentrations of heavy metal ions, such as Cu, Zn, Pb, Cr, Fe, Mo and As, were found in the recovered HS solution. The contents of these heavy metals were below the limits regulated by national standards according to Chinese professional standard NY1110-2010 [21]. Other heavy metal ions, i.e., Cd, Hg, and Mn, and other toxic and harmful substances such as pesticide residues, were not detected. This indicates that the recovered HS solution is safe for the ecosystem with low toxicity, underlining the potential of HS solution recovery by two-stage pilot-scale ED and pilot-scale UF as a water-soluble fertilizer.

Preliminary evaluation of the economic viability of the pilot-scale ED + UF process for sustainable treatment of AIX spent brine was conducted, as shown in Appendix A. It was discovered that the running cost of the pilot-scale ED and pilot-scale UF was 20 and 0.67 Chinese yuan/ton AIX spent brine, respectively. Based on the relevant market value of sodium chloride (700 yuan/ton), the economic earnings of sodium chloride generated by the pilot-scale ED are 31.5 yuan/ton AIX spent brine [3]. This economic benefit could fully neutralize the running cost of pilot-scale ED and pilot-scale UF, and a handsome profit (10.83 Chinese yuan/ton AIX spent brine) could be achieved. Consequently, the pilot-scale ED + UF hybrid process for the AIX spent brine treatment is economically viable and it could be used for practical application.

### 3.5. Pilot Test on the Cultivation of Green Vegetables with HS Liquid Fertilizer

An HS liquid fertilizer was made up by putting nutrient macroelements (ureacontaining nitrogen, and potassium dihydrogen phosphate containing phosphorous, and potassium nitrate containing potassium) into the recovered HS solution. The total content of the macroelement (N + P_2_O_5_ + K_2_O) was 200.1 g/L, and the composition of the HS liquid fertilizer is shown in Appendix A. According to Chinese professional standard NY1106-2010, the constitution of the attained liquid fertilizer can entirely meet the requirements for water-soluble HS fertilizers [21].

To investigate the effectiveness of prepared HS liquid fertilizer compared to conventional fertilizer (Appendix A), the pilot test on the cultivation of green vegetables (Appendix A) was performed. As demonstrated in Figure 7a, the fresh weight of the shoot for green vegetables with HS liquid fertilizer was 48.31 ± 1.1 g/plant, 12.3% higher than that of green vegetables with conventional fertilizer (43.01 ± 1.4 g/plant). Meanwhile, the treated green vegetables with HS liquid fertilizer, compared to conventional fertilizer, showed an increase by 2.6% in the dry weight of the shoot, i.e., increasing from 2.65 ± 0.4 to 2.72 ± 0.6 g/plant. This is because that HSs play a key role in promoting shoot growth and enhancing root initiation in the process of plant metabolism [12]. Otherwise, it is reported that plant growth caused by cell metabolism and increasing nutrient uptake could be boosted by the HSs [45].

As shown in Figure 7b, the plant height of green vegetables increased 5.5% for HS liquid fertilizer compared to conventional fertilizer, suggesting the plant height was stimulated by the HSs. The chlorophyll content of green vegetables with HS liquid fertilizer was 3.2% higher than that of conventional fertilizer, indicating the use of the HSs had a positive effect on the chlorophyll content consistent with the results in the literature [46]. The enhancement of the HSs on photosynthetic pigment (chlorophyll) could be attributed to the increase in carbon dioxide assimilation, photosynthetic rate and Rubisco enzyme activity, leading to the increase of the photosynthetic activity of green vegetables and their yield [47]. Increased chlorophyll pigment could also be due to HS hormonal activity and the increase in nutrient uptake, especially N. This was in agreement with the results of previous reports [45,48]. The effects of HSs on the N and K content of green vegetables were prominent.

The impact of HSs on the N and K content in the green vegetables was significant. As can be seen from Figure 7c, the total nitrogen and total potassium of green vegetables with HS liquid fertilizer were 24.7% and 21.5%, visibly higher than that of green vegetables with conventional fertilizer, respectively, consistent with the results of previous research [45,46]. In addition, HSs could enhance nutrient (N, K) uptake by inducting plasma membrane (PM) H^+^-ATPase activity and increasing root growth in the rhizosphere [48]. It can be seen in Figure 7d that the crude fiber content of the green vegetables increased 5.1% for HS liquid fertilizer compared to conventional fertilizer, while the soluble sugar content was a little lower than that of green vegetables with conventional fertilizer. It is reported the unchanged concentration of total soluble sugars was due to the positive effect of HS on carbon and nitrogen metabolism reflected in sacrificing total soluble sugar and nitrogen to improve protein and starch content [49].

Several studies have explored the relationships between the chemical properties and structure of humic substances and the abilities of HSs to stimulate plant growth and yield improvement [46,50,51,52,53]. Oxygen (O)-containing functional groups (mainly carboxyl and phenolic groups) shown in Figure 2 exist in the structure of HSs, giving it the abilities to form stable chemical bonds with polyvalent metals (M-HA complexes).Stable and soluble complex or natural chelate formation between HS and micronutrients of metallic character, mainly Fe, Cu, Mn and Zn, is beneficial to plant root uptake and improve the plant growth [46,50]. In addition, since the molecular weight of FA is lower than that of HA, the metal complexation increases, leading to higher N content in the leaves of FA-treated plants [46], while the AIX spent brine had more FA than HA [3]. A noticeable increase in the genetic expression of MHa2 isoform (one of two maize PM H^+^-ATPase isoforms: MHa1 and MHa2) was due to the fact that maize seedlings were treated by the low molecular weight HS fraction, increasing PM H^+^-ATPase activity in the root by nutrient root uptake (principally nitrate) and stimulating root growth [51]. Previous studies also indicated that HSs with low molecular size and high content of aromatic, carboxyl and phenolic carbons were particularly active in stimulating plant metabolism. Moreover, the greater the hydrophilic component in the humic substance samples, the higher the activities of HSs on plant physiology, and the more it can promote plant growth [52,53]. In our study, the humic substances in the AIX spent brine had high content oxygen-containing functional groups (carboxyl and phenolic hydroxyl groups), low molecular size and hydrophilic properties [3], thereby promoting the growth and metabolism of green vegetables (Figure 2). Thus, the AIX spent brine-derived HS solution as a water-soluble fertilizer had great potential to cultivate plants in practical applications.

## 4. Conclusions

This work has confirmed the efficacious resource recovery from the AIX spent brine by a pilot-scale ED + UF hybrid process, and the application of recovered NaCl solution as a resin regeneration agent and recovered HS solution as a water-soluble fertilizer for the cultivation of green vegetables. The results have considerable significance for further investigation of electrodialysis and ultrafiltration technologies for sustainable management of the AIX spent brine. The following conclusions can be drawn:AIX spent brine displayed absorption spectrum with typical HS bands, and had a higher proportion of aromatic rings, COO^−^ groups and phenolic OH groups. The apparent molecular weight of HSs in AIX spent brine was located at ~7.2 kDa to ~1.3 kDa.ED can realize the effective separation of NaCl and HSs in AIX spent brine. There covered NaCl solution (≈15% *w*/*w*) could be applied as a regeneration agent of NDMP-3 resins. The NaCl content (0.49% *w*/*w*) in these cond-stage dilute solution can meet the industry standard requirement for the salinity of HS fertilizer.Some of the negatively charged and hydrophilic HSs and BB, low molecular weight and hydrophilic neutrals, and nitrate or nitrite could migrate from the dilute to the concentrate solution. The charge, molecular size, and hydrophilic property of organic matters as well as the ion exchange membrane properties could significantly affect the transport of organic substances during the ED process.The HS content in the retentate solution of UF was >30 g/L, which meets the HS content required of the Chinese professional standard NY1106-2010 for water-soluble fertilizers. The recovered HS solution was safe as a water-soluble fertilizer without phytotoxicity.The recovered AIX spent brine-derived HS liquid fertilizer could improve the growth of the green vegetables. These resulted from that the HSs in the AIX spent brine having high content oxygen-containing functional groups, low molecular size and hydrophilic properties, thus promoting the growth and metabolism of green vegetables.

## Figures and Tables

**Figure 1 membranes-12-00273-f001:**
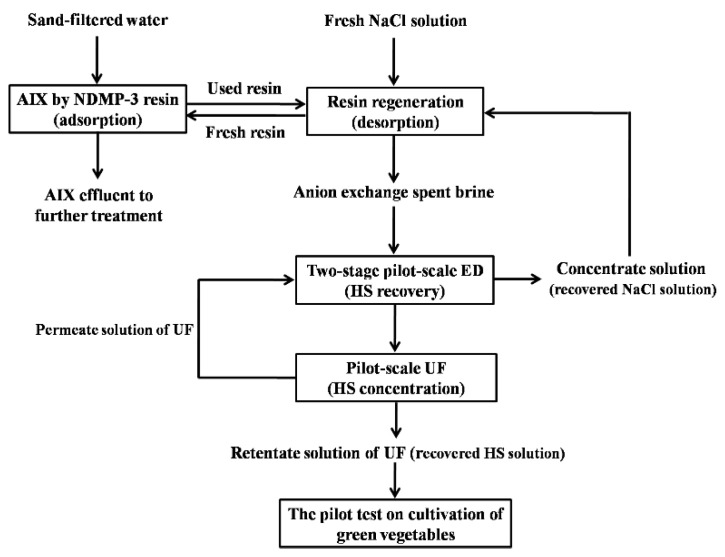
Flowchart of the methodology about sustainable treatment and resource recovery of AIX spent brine applied in this study.

**Figure 2 membranes-12-00273-f002:**
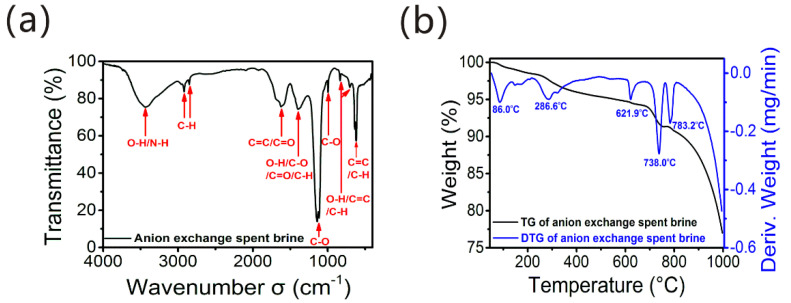
AIX spent brine characterization: (**a**) FT-IR spectrum and (**b**) the TG and DTG profiles.

**Figure 3 membranes-12-00273-f003:**
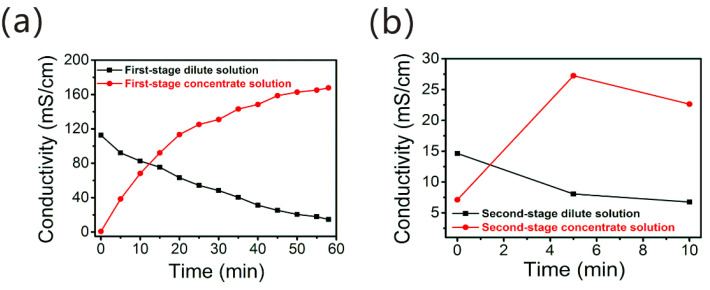
Conductivity changes of AIX spent brine during the first-stage pilot-scale ED (**a**) and second-stage pilot-scale ED (**b**) process.

**Figure 4 membranes-12-00273-f004:**
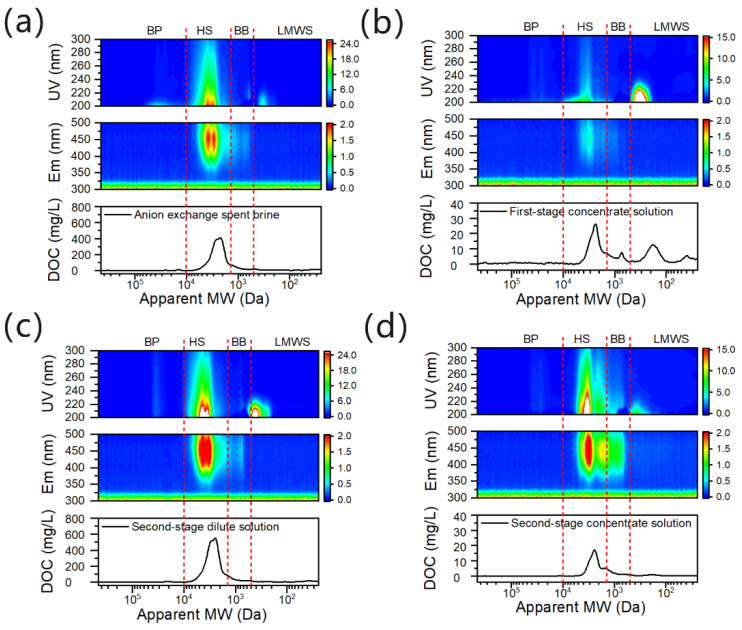
The SEC–DAD–FLD–OCD analyses of AIX spent brine (**a**), first-stage concentrate solution (**b**), second-stage dilute solution (**c**), and second-stage concentrate solution (**d**) in the two-stage pilot-scale ED process.

**Figure 5 membranes-12-00273-f005:**
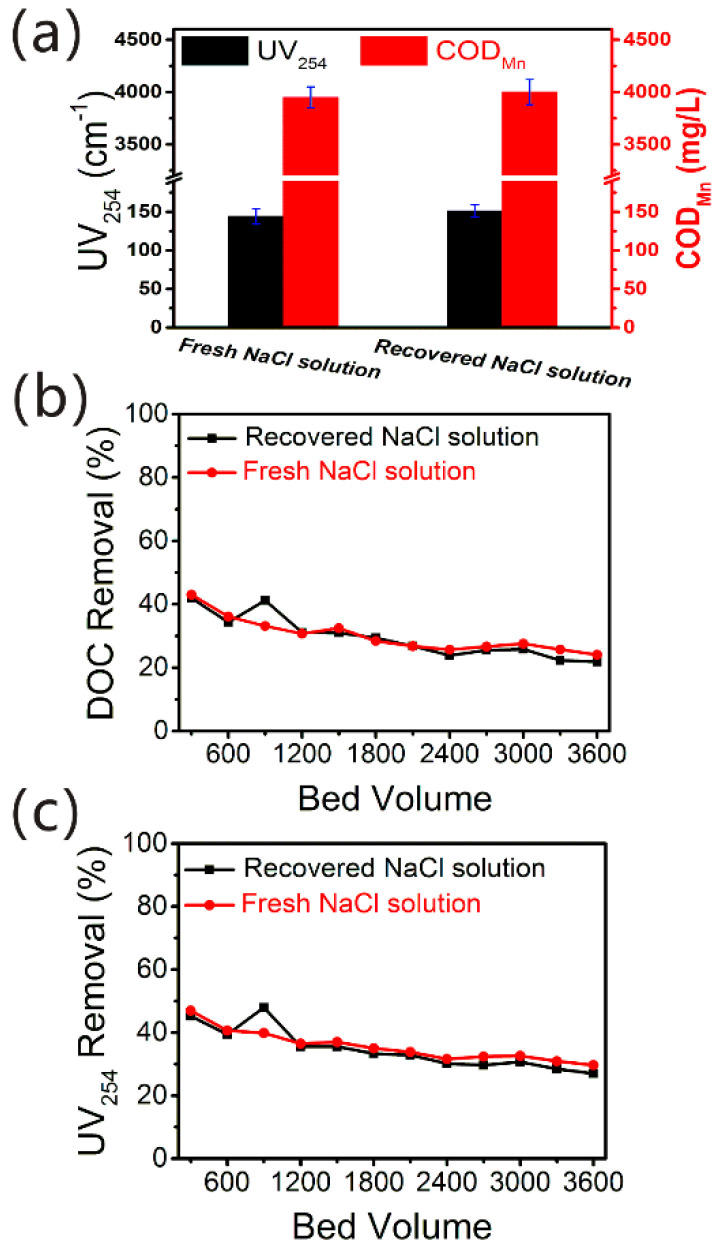
(**a**) The UV_254_ and COD_Mn_ of two AIX spent brine regenerated by recovered NaCl solution and fresh NaCl solution, (**b**,**c**) the removals of UV_254_ absorbance and DOC in sand-filtered water, when NDMP-3 resins regenerated by fresh NaCl solution and recovered NaCl solution were used for readsorption.

**Figure 6 membranes-12-00273-f006:**
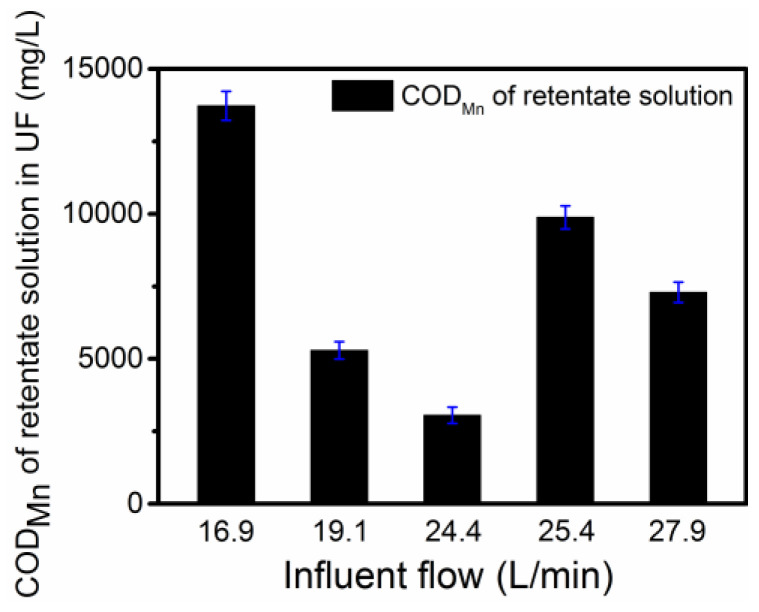
Effect of influent flow on COD_Mn_ of retentate solution in pilot-scale UF.

**Figure 7 membranes-12-00273-f007:**
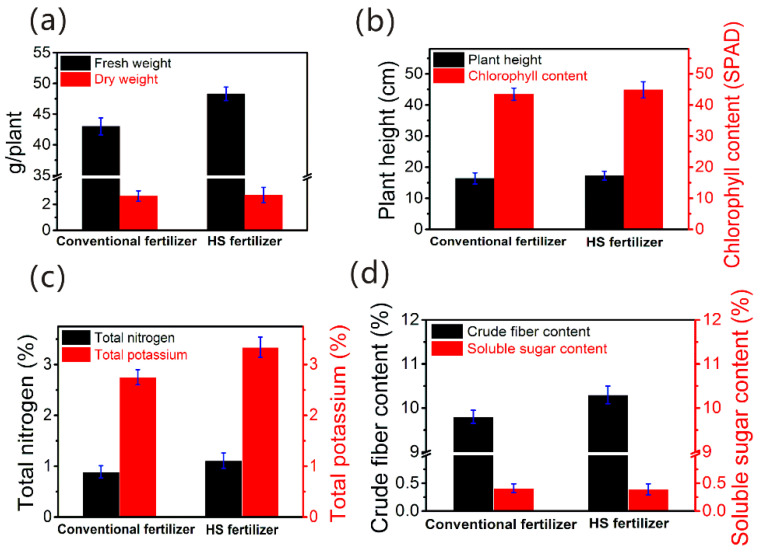
Pilot test on the cultivation of green vegetables with recovered HS solution as liquid fertilizer compared to conventional fertilizer: (**a**) fresh weight and dry weight of the green vegetables., (**b**) plant height and chlorophyll content of the green vegetables, (**c**) total nitrogen and total potassium of the green vegetables, and (**d**) crude fiber content and soluble sugar content of the green vegetables.

**Table 1 membranes-12-00273-t001:** Properties of AIX spent brine, first-stage concentrate solution, first-stage dilute solution, second-stage concentrate solution and second-stage dilute solution during the two-stage pilot-scale ED process.

Parameters	AIX Spent Brine	First-Stage Concentrate Solution	First-Stage Dilute Solution	Second-Stage Concentrate Solution	Second-Stage Dilute Solution
Conductivity (mS/cm)	115.80	167.72	10.82	21.64	6.09
pH	8.27	8.21	8.11	8.25	8.18
COD_Mn_ (mg/L)	2572.3	241.1	4934.2	36.1	5158.7
NaCl content (% *w*/*w*)	9.53	14.92	0.68	1.48	0.49
COD_Mn_ retention (%)	95.5% (total) ^a^	-	95.8%	-	-
Desalination rate (%)	94.7% (total) ^a^	-	90.7%	-	-

^a^ total indicates “Total COD_Mn_ retention” or “Total desalination rate”.

**Table 2 membranes-12-00273-t002:** Characteristics of the recovered HS solution originated from retentate solution in the UF pilot process.

Parmeter	Recovery HS Solution
HS (g/L)	>30
Cu (mg/L)	0.842
Zn (mg/L)	0.920
Cd (mg/L)	ND ^a^
Pb (mg/L)	0.27
Cr (mg/L)	0.71
Hg (μg/L)	ND ^a^
Fe (mg/L)	15.5
Mn (mg/L)	ND ^a^
Mo (μg/L)	532
As (μg/L)	37.6

^a^ ND indicates “Not Detected”.

## Data Availability

Not applicable.

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
