# Peer review of "Sustainable Treatment and Resource Recovery of Anion Exchange Spent Brine by Pilot-Scale Electrodialysis and Ultrafiltration"

_membranes, 2022, doi:10.3390/membranes12030273_

Round 1
Reviewer 1 Report
Hello,
In general, the article is tackling a very important and interesting topic, as it discusses the application of integrated AIX-UF system for resource recovery. The manuscript is well-written and self-explanatory. However, a few comments summarized as follow have to be considered:
- Some proofreading is required.
- I would suggest avoiding long paragraphs (such as that of lines #48-78) and limiting them to 100-200 words while discussing one topic for each paragraph.
- Please use the “,” a thousand commas where required.
- Provide the full name for the used acronyms when mentioned for the first time, like those in line#81.
- In line # 132, the SEC-DAD-FLD-OCD system is explained/discussed/described in the supplementary, but not shown, please amend.
- The conclusion should be made shorter.
Author Response
In general, the article is tackling a very important and interesting topic, as it discusses the application of integrated AIX-UF system for resource recovery. The manuscript is well-written and self-explanatory. However, a few comments summarized as follow have to be considered:
Some proofreading is required.
Q1. I would suggest avoiding long paragraphs (such as that of lines #48-78) and limiting them to 100-200 words while discussing one topic for each paragraph.
Author reply: Thanks for your suggestion. Revisions avoiding long paragraphs have been made in Line #59, Line #206, #228, #290.
Q2. Please use the “,” a thousand commas where required.
Author reply: Agreed. The “,” has been added in Line #54, #82, #131, #148, #151, #186, #200, #203, #210, #212, #219, #224, #225, #229, #236, #240, #241, #256, #322 #333, etc.
Q3. Provide the full name for the used acronyms when mentioned for the first time, like those in line#81.
Author reply: Agreed. Revision has been made in line #80.
Q4. In line # 132, the SEC-DAD-FLD-OCD system is explained/discussed/described in the supplementary, but not shown, please amend.
Author reply: Agreed. Revision has been made in line #129-130.
Q5. The conclusion should be made shorter.
Author reply: Agreed. Revision has been made in Line #564-582.

Reviewer 2 Report
The manuscript submitted by Su et al. presents results on a hybrid treatment option (ED + UF) of an AIX spent brine. The results are interesting and the discussion sound and well presented. Therefore I have just a few suggestions, which, in my opinion, would make this paper even more useful for the readers.
- Line 307 – “the limiting current density was not exceeded” . What was the value of the limiting current density? Please, indicate.
Overall, the authors refer to another paper for the optimization of the ED process parameters, which is understandable, but in order to make the present manuscript more self-consistent, a brief information on the ED operation parameters should be provided at least in the SI file. Namely:
What is the ED operation regime – galvanostatic or potentiostatic? Batch or continuous?
What are the voltage and current values variation for ED unit during the hybrid ED + UF operation ?
- Finally, what are the individual energy consumptions of the ED and UF steps in order to better infer on their relative contributions to the overall energy consumption of the proposed combined treatment option?
Author Response
The manuscript submitted by Su et al. presents results on a hybrid treatment option (ED + UF) of an AIX spent brine. The results are interesting and the discussion sound and well presented. Therefore I have just a few suggestions, which, in my opinion, would make this paper even more useful for the readers.
Q1. Line 307 – “the limiting current density was not exceeded” . What was the value of the limiting current density? Please, indicate.
Author reply: Thanks for your suggestion. When the feed solution contains 0.5% (w/w) NaCl, the limiting current density of TWEDC1 and TWEDA1 ion exchange membranes is > 200 A/m2 in the pilot-scale ED process, which was provided by the manufacturer (Shandong Tianwei Membrane Technology co. Ltd.). The AIX spent brine had 8%-10% (w/w) NaCl, the limiting current density of TWEDC1 and TWEDA1 ion exchange membranes was > 3200 A/m2. In our study, The pilot-scale electrodialysis of AIX spent brine was operated under potentiostatic condition at 70 V, the limiting current density was not exceeded during the ED process.
Q2. Overall, the authors refer to another paper for the optimization of the ED process parameters, which is understandable, but in order to make the present manuscript more self-consistent, a brief information on the ED operation parameters should be provided at least in the SI file. Namely:
What is the ED operation regime – galvanostatic or potentiostatic? Batch or continuous?
Author reply: Thanks for your comments. The pilot-scale electrodialysis of AIX spent brine was carried out under potentiostatic condition at 70 V. And the batch operating mode was used. This is because pilot-scale electrodialysis was used to treat AIX spent brine when it accumulated to a certain amount (>5 tons) in the tank of AIX spent brine. When the NDMP-3 anion exchange resin adsorption was conducted for a week, the resins were saturated, and only 1 ton of AIX spent brine was produced when the resins are regenerated with NaCl. Revision has been made in Revised Supplementary Material.
Q3. What are the voltage and current values variation for ED unit during the hybrid ED + UF operation ?
Author reply: Thanks for your suggestion. The voltage and current value variation for pilot-scale ED unit was shown in Table 1:
Table 1 Current and voltage versus time of two-stage pilot-scale ED.
|
First-stage pilot-scale ED: |
||
|
t/min |
U/V |
I/A |
|
0 |
70 |
48 |
|
5 |
46 |
100 |
|
10 |
41 |
100 |
|
15 |
42 |
100 |
|
20 |
60 |
100 |
|
25 |
61 |
100 |
|
30 |
61 |
100 |
|
35 |
45 |
100 |
|
40 |
44 |
100 |
|
45 |
36 |
100 |
|
50 |
44 |
100 |
|
55 |
70 |
51 |
|
58 |
70 |
37 |
|
Second-stage pilot-scale ED: |
||
|
t/min |
U/V |
I/A |
|
0 |
70 |
42 |
|
5 |
70 |
6 |
|
10 |
70 |
3 |
As can be seen from Table 1, the first-stage pilot-scale electrodialysis system has experienced an operating state of constant voltage (70 V) first, and then constant current (100 A), and last constant voltage (70 V). The second-stage pilot-scale electrodialysis was operated at constant voltage (70 V) until the current could no longer drop, that is, the anion exchange spent brine could no longer be desalinated.
Q4. Finally, what are the individual energy consumptions of the ED and UF steps in order to better infer on their relative contributions to the overall energy consumption of the proposed combined treatment option?
Author reply: Thanks for your comments. The running cost of sustainable treatment of AIX spent brine was shown in Table S4 in Revised Supplementary Material.
Table S4 Running cost of sustainable treatment of AIX spent brine.
|
Treatment processes |
Operating cost (Yuan/ton) |
|
ED pilot |
20 |
|
UF pilot |
0.67 |
|
NaCl recovery |
-31.5a |
|
Total cost |
-10.83a |
aNegative sign indicates profit.
As can be seen from Table S4, the energy consumption (running cost) of pilot-scale ED and pilot-scale UF was 20 and 0.67 Chinese Yuan/ton AIX spent brine, respectively.
We tried our best to improve the manuscript and made some changes in the manuscript. These changes will not influence the content and framework of the paper. And here we did not list the changes but marked in red in revised paper.
We appreciate for Editors/Reviewers’ warm work earnestly, and hope that the correction will meet with approval.
Once again, thank you very much for your comments and suggestions.